# Lateralized and Segmental Overgrowth in Children

**DOI:** 10.3390/cancers13246166

**Published:** 2021-12-07

**Authors:** Alessandro Mussa, Diana Carli, Simona Cardaropoli, Giovanni Battista Ferrero, Nicoletta Resta

**Affiliations:** 1Department of Public Health and Pediatric Sciences, University of Torino, 10126 Torino, Italy; diana.carli@unito.it (D.C.); simona.cardaropoli@unito.it (S.C.); 2Pediatric Clinical Genetics Unit, Regina Margherita Children’s Hospital, Città della Salute e della Scienza di Torino, 10126 Torino, Italy; 3Pediatric Onco-Hematology, Stem Cell Transplantation and Cell Therapy Division, Regina Margherita Children’s Hospital, Città della Salute e della Scienza di Torino, 10126 Torino, Italy; 4Department of Clinical and Biological Sciences, University of Torino, 10124 Torino, Italy; giovannibattista.ferrero@unito.it; 5Department of Biomedical Sciences and Human Oncology (DIMO), Medical Genetics, University of Bari “Aldo Moro”, 70121 Bari, Italy; nicoletta.resta@uniba.it

**Keywords:** overgrowth, lateralized, segmental, Beckwith–Wiedemann, PIK3CA, somatic, mosaic, PROS, cancer screening, cancer predisposition

## Abstract

**Simple Summary:**

Congenital lateralized or segmental overgrowth (LO) disorders are conditions characterized by excessive tissue growth of a body region often associated with a predisposition to cancer. LOs are caused by mosaic DNA anomalies, that is, they are present only in a part of the cells making up the body. LOs have an extremely heterogeneous clinical presentation: they widely overlap in presentation, are difficult to frame from a clinical point of view and have a diagnostic complexity representing a challenge for the clinician who approaches them. Here we review the key features of the various LOs, expose their molecular causes, and detail the implications for each of them, such as the need for specific cancer screening or the possibility of treatment. The latter represents a recent scientific achievement in medicine, allowed by the development of precision drugs finely tuning cellular pathways involved in growth and tumorigenesis deranged in LO.

**Abstract:**

Congenital disorders of lateralized or segmental overgrowth (LO) are heterogeneous conditions with increased tissue growth in a body region. LO can affect every region, be localized or extensive, involve one or several embryonic tissues, showing variable severity, from mild forms with minor body asymmetry to severe ones with progressive tissue growth and related relevant complications. Recently, next-generation sequencing approaches have increased the knowledge on the molecular defects in LO, allowing classifying them based on the deranged cellular signaling pathway. LO is caused by either genetic or epigenetic somatic anomalies affecting cell proliferation. Most LOs are classifiable in the Beckwith–Wiedemann spectrum (BWSp), PI3KCA/AKT-related overgrowth spectrum (PROS/AROS), mosaic RASopathies, PTEN Hamartoma Tumor Syndrome, mosaic activating variants in angiogenesis pathways, and isolated LO (ILO). These disorders overlap over common phenotypes, making their appraisal and distinction challenging. The latter is crucial, as specific management strategies are key: some LO is associated with increased cancer risk making imperative tumor screening since childhood. Interestingly, some LO shares molecular mechanisms with cancer: recent advances in tumor biological pathway druggability and growth downregulation offer new avenues for the treatment of the most severe and complicated LO.

## 1. Introduction

Lateralized overgrowth (LO), or segmental overgrowth, is defined as an increase in tissue growth of various origins (skeletal, muscular, fibrous, vascular, adipose, or any association of these) in any region of the body [1]. LO and segmental overgrowth are used synonymously and currently replace the previous nomenclature. The latter largely referred to body/hemi hyperplasia/hypertrophy, terms that implicated the histologic definition of cell proliferation or increased cell size, respectively. LO often leads to body asymmetry caused by an increase in the length/circumference of a limb, or in body region volume. Asymmetry is generally considered pathological if it exceeds at least 10% compared to the contralateral area. This threshold is theoretical and underlines that it is considered physiological to have minor asymmetries in certain areas of the body: an example is represented by the very common clinical finding of discrepancy in the length of the lower limbs of less than one centimeter that occurs in many adults. Indeed, there is no need for imaging studies to define LO: LOs are subjective and evident at the clinical evaluation (i.e., “from the end of the bed”). Objective methods, such as tape measurements or imaging studies of both soft and skeletal tissue by computed tomography, magnetic resonance, 3D technologies or DEXA scanning can be useful to better define, characterize and quantify the degree of LO and in the differential diagnosis of such overgrowth disorders [2]. However, these measurements are not standardized as yet [3].

LO may be caused by genetic or epigenetic defects leading to disruption in cell growth and over signaling in cellular proliferation pathways resulting in tissue overgrowth and cell proliferation. LO can be part of a syndrome or isolated. It is a cardinal feature of several overgrowth disorders as with the Beckwith–Wiedemann spectrum (BWSp), Proteus syndrome and many of the clinical entities related to somatic pathogenic variants in *PIK3CA* (PIK3CA-Related Overgrowth Spectrum, PROS). LO can less often be seen also in the context of several other overgrowth disorders [4,5], such as mosaic RASopathies, neurofibromatosis type I, diffuse capillary malformation with overgrowth (DCMO), and *PTEN* hamartomatous tumor syndrome (PHTS). Contrarily, Isolated Lateralized Overgrowth” (ILO)—previously defined Isolated hemi-hyperplasia (OMIM # 235000)—is not associated with other developmental anomalies [3]. In both its isolated and syndromic form, LO is frequently associated with cancer predisposition. The objective of this review is to provide an overview of LO molecular basis, diagnostic-care protocols, cancer screening and new drug treatment strategies currently under study.

## 2. Beckwith–Wiedemann Spectrum (BWSp)

### 2.1. Overview

BWSp (OMIM # 130650) is the most common congenital overgrowth disorder with a prevalence approaching 1:10,000 live births in the general population [6] and of 1:1100 in children conceived by assisted reproduction techniques [7]. Its clinical features include fetal macrosomia, postnatal overgrowth, abdominal wall defects, LO, macroglossia, auricular anomalies (indentations in the ear lobes and helix of the pavilion), organomegaly, nephro-ureteral malformations, hyperinsulinism with hypoglycemia, and oncological predisposition. The malformations can be variably associated and have different degrees of severity, thus resulting in a widely variable clinical spectrum. LO in BWSp can rarely be identified prenatally, not even in particularly severe forms [8,9,10,11]. BWSp diagnosis is clinical and based on criteria that have been refined over time and recently reviewed by leading experts in pathology [12] (Table 1). BWSp includes typical Beckwith–Wiedemann syndrome (BWS) cases (when several of the diagnostic criteria are present with a score ≥ 4 points, with or without associated genetic anomalies), atypical BWS (when typical molecular anomalies are found in patients with few diagnostic criteria and score is <4 points), and cases with ILO and associated molecular BWSp abnormalities (Figure 1) [13].

### 2.2. Molecular Basis of the BWS Spectrum

The molecular basis of BWSs is complex and heterogeneous and represents a model of genomic imprinting defects. While most human genes are expressed by both maternal and paternal alleles, genes subject to genomic imprinting are expressed monoallelically based on parental origin. This differential expression mechanism is regulated epigenetically by the differential methylation of parental alleles at the differentially methylated regions (DMR) of two imprinting centers (ICs) at 11p15.5 in BWSp. IC1 and IC2 regulate the transcription of two clusters of genes involved in cell cycle progression. Epigenetic anomalies of these regions are found in nearly 85% of cases of BWSp. Figure 2 details the molecular mechanisms of the syndrome: (a) Loss-of-Methylation (LoM) in IC2 occurring in more than 50% of cases, (b) Gain-of-Methylation (GoM) in IC1 detected in 5–10% of cases, (c) paternal uniparental disomy of chromosome 11 (UPD(11)pat) identified in 20–25% of cases, (d) inactivating mutations of the *CDKN1C* gene, observed in 5% of cases, and responsible for about half of heritable cases [12,14].

### 2.3. Lateralized Overgrowth in BWSp

LO is one of the cardinal features of BWSp and mild BWSp can be evident with ILO (Figure 3). LO is present in approximately one-third of BWSp patients. Although LO can be found in patients with each of the molecular subtypes of BWSp [15], this feature shows a strong association with UPD(11)pat: in this group, its prevalence is around 60–80%, while it is less common in the IC2-LoM (25%) and the IC1-GoM (35%) groups, and very rare in *CDKN1C* mutation cases (less than 5% of cases) [16,17]. LO is the feature of BWSp showing the strongest association with cancer development [17,18,19,20]. LO has a mild trend in worsening during growth, especially if compared to other conditions with segmental overgrowth [21]. Mainly the more severe LLD tend to have worse evolution. Despite orthopedic interventions and epiphysiodesis being effective in correcting the LLD, the LO and LLD remain a significant cause of disability and morbidity in adulthood [22].

### 2.4. Cancer Screening

Cancer surveillance represents the most delicate and controversial aspect of the clinical management of patients with BWSp. The (epi)genotype-phenotype correlation studies allowed us to stratify the oncological risk with respect to the different molecular subgroups. For the majority of patients, those affected by IC2-LoM, the tumor risk is low (about 2%) and mostly encompasses hepatoblastoma or rhabdomyosarcoma. On the opposite, the risk is relevant in patients with UPD(11)pat (about 15%) and these patients develop any kind of tumor seen in BWSp, mainly Wilms tumor (WT) and hepatoblastoma, but also adrenal carcinoma and pheochromocytoma. The highest risk is seen in patients with IC1-GoM and approaches 25%, almost exclusively for WT [23,24].

The cornerstones of surveillance in BWSp are abdominal ultrasound (US) for early detection of WT [25] and the serum alpha-fetoprotein assay [26,27,28] used for early diagnosis of hepatoblastoma. The US can detect also hepatoblastoma at advanced stages with respect to alpha-fetoprotein [29,30], as well as other rarer abdominal neoplasms. The US up to 8 years of age and alpha-fetoprotein assay up to 4 years of age are repeated quarterly based on tumor growth data [31]. There is a currently unanimous consensus on performing quarterly ultrasound (US) WT screening up to the eighth year of age in patients with IC1-GoM, UPD(11)pat and negative molecular tests [31]. The position of experts on US screening in patients with IC2-LoM and on screening for hepatoblastoma by means of the alpha-fetoprotein assay, on the other hand, appears varied [31,32]. In many European countries, the IC2-LoM group is not screened as it has almost no risk for WT and alpha-fetoprotein screening is not adopted given the overall low-risk of hepatoblastoma (less than 5%) [31]. On the other hand, in the USA, adopting a lower risk threshold (~1%) the IC2-LoM group is screened and the alpha-fetoprotein assay screening is suggested [33]. However, epidemiology suggests that most tumors in patients with IC2-LoM and hepatoblastoma cases occur within the first 2 years of life, therefore, we adopt the US in IC2-LoM and alphafetoprotein assay every 3 months up to 24–30 months of age for all patients [34]. A rare epigenetic subtype of BWSp, androgenetic chimerism, is characterized by severe features and very high cancer risk and needs a more frequent and intense screening protocol that can be extended to young adulthood [35].

Management of the adrenal masses in children with BWSp has been reviewed by MacFarland et al. [36].

It is common that children with WT are diagnosed within the BWSp secondarily to cancer diagnosis [37]. Moreover, it has been recently demonstrated that up to one-third of children with WT or hepatoblastoma have a 11p15.5 epimutation detectable in the blood [38] and suggested that this testing might be included among those by a sequencing-based approach to screen for a cancer predisposition in children with such tumors [38].

## 3. PIK3CA-Related Overgrowth Spectrum (PROS)

### 3.1. Overview

The acronym PROS (*PIK3CA*-Related Overgrowth Spectrum) identifies a group of developmental disorders characterized by segmental overgrowth caused by activating mutations of the catalytic subunit phosphatidylinositol-4,5-bisphosphate 3-kinase alpha (*PIK3CA*). PROS is a generic nosological definition, introduced in 2015, to group different clinical entities, independently described before it was discovered that *PIK3CA* mutations were their cause [39,40]. *PIK3CA* pathogenic variants at the base of the clinical pictures of PROS are commonly postzygotic, coming to configure a picture of tissue mosaicism [41]. The extension of the mosaicism and, to a lesser extent, the genotype is responsible for the different phenotypes of the PROS [39,40,42] (Figure 4). The degree of mosaicism is determined by the timing of embryonic development in which the DNA mutates, determining the body extension and the kind of tissues involved. The severity in terms of the progression of the tissue overgrowth phenotype is determined by the degree of hyperactivation of the cascade resulting from the pathogenic variant [40]. These variables cause a high degree of interindividual phenotypic heterogeneity within the clinical spectrum, which—also on the basis of the definitions for some time in use—can schematically include these distinct entities:Type I macrodactyly [43];Muscle fibrous hyperplasia (MH, FH) [44] associated or not with vascular anomalies (fibroadipose vascular anomalies, FAVA) or adipose overgrowth (fibroadipose overgrowth, FAO) including hemihyperplasia multiple lipomatosis (HHML) characterized by overgrowth of the entire hemi-body and multiple lipomas;Facial Infiltrating Lipomatosis (FIL) is characterized by overgrowth mostly involving adipose and soft tissue localized to the facial district with a clear-cut asymmetry of the face;Epidermal nevi (EN), Seborrheic Keratosis (SK) and benign lichenoid keratosis (BLK);Congenital lipomatous overgrowth, vascular malformations, epidermal nevi, scoliosis/skeletal and spinal syndrome (CLOVES, OMIM # 612918) with or without acral deformities as large, wide feet and hands, macrodactyly, and sandal gap [45];Klippel–Trenaunay syndrome (KTS, OMIM # 149000) is characterized by disproportionate growth of soft and bony tissue of a body region (typically involving the lower extremities) and combined with cutaneous capillary, lymphatic, and venous malformations consisting in varicosity, the persistence of embryonal veins, or a lymphatic component [46];Megalencephaly-capillary malformation polymicrogyria syndrome (OMIM # 602501, MCAP), characterized by macrocephaly, capillary malformation, somatic overgrowth, body asymmetry, brain anomalies and progressive megalencephaly, and neurodevelopmental delay [47];Hemimegalencephaly (H-MEG) and dysplastic megalencephaly (D-MEG) characterized by cerebral overgrowth and a variable degree of cortical development malformation [48];Capillary malformation of the lower lip, lymphatic malformation of the face and neck, asymmetry and partial/generalized overgrowth (CLAPO, OMIM # 613089) [49].

Some of the entities included in the PROS spectrum can be diagnosed clinically thanks to pathognomonic signs or by characteristic body location. However, there is a significant phenotypic overlap between the various forms for which the proposed classification is difficult to apply. Clinical criteria have, therefore, been studied for a clinical diagnosis. The diagnosis of PROS spectrum entities is based on four clinical criteria [39]:
Finding of somatic mutation in the *PIK3CA* gene;Congenital or early childhood clinical presentation;Localized growth trend with a mosaic pattern;Presence of at least a or b criteria:
(a)Two or three of the following anomalies:i.Overgrowth: adipose, muscular, nervous, skeletalii.Vascular malformations: capillary, venous, arteriovenous, lymphaticiii.Epidermal nevus(b)Presence of at least one of the following isolated anomalies:
iv.Extensive lymphatic malformationv.Macrodactyly or acral overgrowth (hands, feet, limb)vi.Fat overgrowth localized to the trunkvii.Hemimegalencephaly/dysplastic megalencephaly/focal cortical dysplasiaviii.Epidermal nevusix.Seborrheic keratosisx.Benign lichenoid keratosis


### 3.2. Molecular Diagnosis

The pathogenetic variants of the *PIK3CA* gene are commonly somatic hampering the molecular assay on DNA extracted from blood leukocytes: it is usually needed testing DNA from resected tissues or elective skin/other tissue biopsies. It is advisable that the analysis is conducted on a fresh sample and not on cultured tissue. A molecular diagnosis requires a careful selection of the area of tissue on which a biopsy should be performed and a high sensitivity analytical method for the detection of variants with a low degree of tissue mosaicism. Although numerous *PIK3CA*-activating variants have been identified in cancer, three mutational hotspots are characterized by strong oncogenic activity: p.His1047Arg, p.Glu542Lys, and p.Glu545Lys account for 80% of the spectrum [50]. Rapid screening for recurrent hotspot mutations may represent a reasonable first-tier diagnostic approach, to be completed, if negative, with the sequencing of the exons of the *PIK3CA* gene. A high-depth NGS approach is preferable as more sensitive, allowing for the detection of Variant Allele Frequencies (VAFs) of 1–3% magnitude, compared to the Sanger method that has rather limited sensitivity and is not commonly able to detect VAFs lower than 15–20%.

A blunt correlation between genotype and phenotype is described. The variants that cause MCAP and other phenotypes with central nervous system involvement are often weakly activating and variously distributed within the PIK3CA gene, while the mutations with a stronger degree of activation characterize the more localized forms of PROS—for example, CLOVES syndrome and dysplastic megalencephaly—are mainly localized in mutational hotspots frequently mutated in sporadic neoplasms [40].

### 3.3. Pathophysiology and Druggability

The PIK3CA gene encodes the 110 kD catalytic subunit of PI3K, which converts phosphatidylinositol (3,4) bisphosphate (PIP2) to phosphatidylinositol (3,4,5) phosphate (PIP3) upon stimulation by a growth factor that binds to a tyrosine kinase receptor. PIP3 causes phosphorylation and translocation of PDK1 to the cell membrane, which in turn phosphorylates the serine/threonine kinase AKT, which stimulates cell proliferation through the mTORC1 complex-mediated signal transduction cascade. The heterozygous activating variants in PIK3CA lead to a hyperactivation of the PI3K/AKT/mTOR cascade resulting in overgrowth. The PI3K/AKT/mTOR signaling pathway is among the most dysregulated pathways described in human cancers. *PIK3CA* is indeed most frequently mutated in breast cancer (28.8%), principally in estrogen receptor (ER)-positive carcinomas (38.9%), as well as endometrium (27.4%) and urinary tract (20.2%) cancers [51].

To get a mechanistic understanding of PROS pathophysiology and to test treatments, faithful PROS experimental models are needed, ideally spanning the entire mutational spectrum. Temporarily regulated cell type-specific production of mutant *PIK3CA* alleles in animal models, in conjunction with the use of modified human stem cells and organoids, would be a useful tool to study LO in PROS. Several mice models for preclinical research demonstrated the complexities in modeling mosaic disorders due to the heterogeneity in mutational time and location in embryos. Mice constitutively expressing mutant allele Pik3ca-H1047R (one of the most oncogenic mutations) die around E9.5–E10.5 supporting the theory that strongly activating variants are lethal in the germline status. Inducible CreER expression driven by the T-Brachyury promoter can overcome early lethality and mimic PROS: specifically, mice with different mosaicism levels developed vascular anomalies similar to those observed in humans [52]. Several studies modeling Pik3ca-driven brain overgrowth in mice showed the relevance of mutation timing and cellular context, confirming that brain overgrowth depends on p110a activation and that postnatal Pik3ca-E545K induction cause neurological impairment [53]. Furthermore, biological models support the existence of a threshold beyond which carcinogenic mechanisms are triggered, as observed in the dosage dependent cellular consequences of homozygous or heterozygous H1047R variant [50,54]. Interestingly, the differential engineering of isogenic human induced pluripotent stem cells with the heterozygous or homozygous knocking of PIK3CA H1047R, led to different cellular behavior: *wt*-like in the heterozygous state, cancer-like (with transcriptional remodeling, loss of typical epithelial morphology, decreased differentiation, up-regulation of stemness markers) in the homozygous one.

These observations led to extensive research in drugs inhibiting the PI3K-mediated signaling pathway at various levels: dozens of PI3K/AKT/mTOR pathway inhibitors are in clinical development and few have been approved for clinical use [55]. The oldest one is rapamycin, which has been extensively used in PROS [56].

More selective drugs have been recently introduced, as PI3K and AKT inhibitors. The AKT-inhibitor Miransertib is a novel, orally available, selective pan-AKT inhibitor with proven in vitro efficacy used in AKT-mutated cancers and Proteus syndrome, with promising effectiveness in PROS [57]. PI3K inhibitors can be subdivided into pan-PI3K inhibitors, isoform-selective PI3K inhibitors, and dual PI3K/mTOR inhibitors [58]. Most pan-PI3K inhibitors have been discontinued owing to insufficient efficacy and toxicities. Alpelisib (BYL719, Piqray^®^) currently represents the most promising treatment and for which drug trials are underway [59,60]. Preliminary data concerning the treatment in the context of off-label administration or compassionate use are available [60,61].

### 3.4. PROS and Cancer

PROS patients require a multidisciplinary and personalized approach based on the specific systems and organs involved and include pediatricians, geneticists, neurologists, dermatologists, orthopedists, plastic surgeons, neurosurgeons, otolaryngologists. Although activating variants of *PIK3CA* are frequent in some tumors of different tissue origin, at present, there is insufficient evidence to consider PROS with increased cancer risk. Rare neoplasms have been described in some patients: Wilms tumor seems to be the most common tumor type in PROS patients (occurring in nearly 3.3% of patients with CLOVES) [62]. Although the risk appears to be lower overall than many other overgrowth syndromes, it is discussed whether to indicate a surveillance protocol based on quarterly abdominal US. On this issue, further investigations into cancer risk are obviously needed.

## 4. Proteus Syndrome and AKT1-Related Overgrowth Spectrum

Proteus syndrome is a progressive segmental overgrowth disorder characterized by a wide range of severity, varying greatly from patient to patient, as observed in the clinically overlapping disorders of the PROS. Excessive growth is typically irregular, asymmetric and disproportionate, progressive and severe causing disfigurement and body deformities leading to variable symptoms and complications. Overgrowth can virtually affect any tissue, the most commonly affected is the skeleton, connective tissue, skin, fat, and central nervous system and internal organs. Usually, there are few or no signs at birth: affected tissue overgrowth usually develops and progresses quickly beginning in the first childhood and persistently progressing during life.

Proteus syndrome is associated with a range of tumors (meningiomas, ovarian cystadenomas, breast cancer, parotid monomorphic adenoma, mesothelioma) and predisposition to complications (deep vein thrombosis, pulmonary embolism, development of pulmonary bullae and cysts). Typical of Proteus syndrome is cerebriform connective tissue nevi (characterized by deep grooves and gyrations), linear verrucous epidermal nevus (streaky, pigmented, rough nevus that often follows the lines of Blaschko), capillary, venous, or lymphatic vascular malformations, and fat tissue dysregulation with alternations of lipomatous overgrowth and lipoatrophy.

The variety of affected tissue and the clinical overlap with other somatic overgrowth disorders makes the diagnosis challenging. A clinical criteria scoring system and specific molecular test lead to the diagnoses of classic Proteus syndrome or AKT1-related overgrowth spectrum when a variant is found in a patient with few of the typical features [63]. Proteus syndrome has been almost exclusively associated with the mosaic c.49G > A p.(Glu17Lys) pathogenic gain-of-function variant in *AKT1* [64], also observed in cancers [51,65].

Hyperactivation of *AKT1* is responsible for over signaling through the PI3K/AKT/mTOR pathway, a key regulator of cell proliferation and survival [51] and represents an attractive target for therapeutic intervention [51]. Miransertib (ARQ 092) is now being evaluated for the treatment [66,67].

## 5. PTEN Hamartoma Tumor Syndrome (PHTS)

*PTEN* hamartoma tumor syndrome (PHTS) is an autosomal dominant condition characterized by intellectual disability, overgrowth, and tumor predisposition including the overlapping phenotypes of Cowden, Bannayan-Riley-Ruvalcaba, *PTEN*-related Proteus and Proteus-like syndromes [68].

Cowden syndrome is multiple hamartomatous and typically manifests in early adulthood and is characterized by macrocephaly, trichilemmomas, papillomatous papules and increased risk for benign and malignant tumors. Bannayan-Riley-Ruvalcaba syndrome is considered the pediatric corresponding of Cowden syndrome: it is a congenital disorder with generalized/lateralized overgrowth with macrocephaly, developmental delay/autism, proximal myopathy, intestinal hamartomatous polyposis, lipomas, scoliosis and pigmented macules of the glans penis. PTEN-related Proteus and Proteus-like syndromes are characterized by progressive segmental overgrowth of several tissues including the skeleton, skin, adipose tissue, and central nervous system, associated with complications and vein thrombosis [69]. LO is typically seen in Bannayan-Riley-Ruvalcaba syndrome and *PTEN*-related Proteus and Proteus-like syndromes. Some PHTS patients with localized overgrowth phenotypes have been described with type 2 segmental mosaicism consisting of anatomically localized coexistence of a postzygotic variant in *PTEN in trans* to the germline *PTEN* mutant allele, resulting in biallelic mutations in some cell lines. This results in a clinical phenotype characterized by a more severe segment of disease in a patient with otherwise typical disease distribution defined as *PTEN*-related Proteus syndrome or SOLAMEN (segmental overgrowth, lipomatosis, arteriovenous malformation and epidermal nevus) [70,71,72].

Diagnostic criteria for PHTS are available in adults and children [73]. PHTS shows a high risk for several tumors, including the thyroid, bowel, breast, endometrium, and kidney. Tumorigenesis originates from the inactivation of the second (wild-type) allele via a second-hit somatic mutation [74,75]. Specific recommendations for cancer screening in youth and adulthood exist [76,77].

The diagnosis of PHTS is established upon the identification of impaired transcription of *PTEN*: this can result from either a pathogenic germline variant or deletion/duplication in the gene or its promoter, detectable by gene-targeted sequence analysis or copy number variants testing. *PTEN* is a tumor suppressor gene inhibiting the PI3K/AKT/mTOR signaling cascade: its loss of function results in a dysregulation of this pathway [73]. This represents a rational therapeutic target as oversignaling in the cascade can be mitigated by mTOR inhibitors: sirolimus or everolimus have been shown effective in ameliorating PHTS features and symptoms [78,79,80].

## 6. Mosaic RASopathies

Mosaic RASopathies are a group of diseases caused by mosaic pathogenetic variants in genes of the RAS/MAPK signaling pathway and characterized by a wide spectrum of segmental dysplasia and overgrowth with different involvement of eye, skin, heart, brain, skeletal, and soft tissues. Mosaic RASopathies encompass oculoectodermal syndrome (OES) (OMIM 600268), encephalo-cranio-cutaneous lipomatosis (ECCL) (OMIM 613001), Schimmelpenning-Feuerstein-Mims syndrome (SFMS) (OMIM 163200), RAS-associated autoimmune lymphoproliferative syndrome (OMIM 614470), sebaceous and epidermal nevus (OMIM 162900), low-flow vascular malformations and AVMs (OMIM 108010), melorheostosis (OMIM 155950), cutaneous-skeletal hypophosphatemia (CSHS), and melanocytic nevus syndrome (OMIM 137550) [81,82,83].

The main clinical features include vascular malformations, congenital nevi, eye, and brain anomalies. Segmental overgrowth is common as well as embryonal tumors have been described in the mosaic RASopathies [81,84], thus enclosing the mosaic RASopathies in the differential diagnosis of LO [83,84,85]. The phenotype is variable due to the timing of the mutational event during embryonal life, level of RAS/MAPK pathway over signaling, and body tissue distribution of the pathogenic variant [81].

To date, the genes associated with mosaic RASopathies are *BRAF*, *FGFR1*, *HRAS*, *KRAS*, *MAP2K1*, *NF1*, *NRAS*, *RASA1*, but other genes of the pathway will be likely associated in the near future. The spectrum of pathogenic variants in these genes in human cancer and in mosaic RASopathies differs from the mutation spectrum observed in germline RASopathies, suggesting their lethality in the germline form and explaining the markedly different phenotype of mosaic RASopathies when compared to the germline ones [81,85].

Downregulation of the RAS/MAPK over signaling with targeted medical therapies, such as MEK inhibitors, is a recently introduced promising therapeutic option for these patients [83,84].

## 7. Lateralized Overgrowth Syndromes with Vascular Anomalies

Diffuse capillary malformation with overgrowth (DCMO) is a clinical diagnosis identifying patients with multiple and extensive capillary malformations (CM) associated with overgrowth of a body segment, usually an entire hemisome or limb. CM in DCMO are typically reticulate, pale, extensive, and diffuse, characterized by multiple anatomic regions contiguously stained [86]. The absence of deep venous varicosities, the persistence of embryonic vessels or lymphatic components differentiate DCMO from KTS, together with being virtually free of major complications and limited evolution over time. DCMO is an entity recently described: confusion still exists in the clinical definition of many conditions characterized by overgrowth and vascular anomalies and many of such conditions are identified as yet with definition and as syndromes dating back to the pre-NGS era. Currently, the classification of vascular anomalies with overgrowth is rapidly evolving thanks to massively parallel sequencing. Somatic variants in *GNAQ* and *GNA11* have been found in DCMO as well as in overlapping disorders, such as Sturge–Weber syndrome [87]. More recently, it has been highlighted that DCMO can also be caused by pathogenetic variants in *PIK3CA*: molecular testing is, therefore, necessary to correctly categorize and characterize cases of DCMO.

## 8. Isolated Lateralized Overgrowth (Not Belonging to Other Overgrowth Disorders)

### 8.1. Overview

The diagnosis of ILO is made clinically when segmental overgrowth is not attributable to a specific multiple developmental disorder and shows no associated anomaly or malformation. Diagnosing this clinical entity can be tricky for two reasons. First, it is not trivial to distinguish overgrowth of a region of the body from contralateral hypoplasia/hypotrophy, both because the mechanisms involved in the pathogenesis are different, and because of the profound differences in clinical management. An example is represented by segmental undergrowth/hemyhypoplasia that can be seen as an isolated feature in mild forms of Silver–Russell syndrome [88]. The latter is the mirror phenotype of BWS and is caused by the opposite (epi)genetic changes of BWS [89]. ILO, therefore, implies the differential diagnosis with causes of isolated undergrowth of a body area [90], as it is not easy to distinguish which of the body part is normal and which is affected by an over/under-growth condition. In this view also acquired congenital asymmetries, not of genetic origin must be considered: examples are vascular disruption during fetal life or iatrogenic limb hypoplasia consequent to vascular damage following central lines positioning. Correctly framing the clinical presentation is key to requesting appropriate molecular tests and setting up a correct management approach and follow-up.

### 8.2. Molecular Bases

The etiopathogenetic mechanisms at the basis of ILO include somatic mosaicisms for genetic or epigenetic mutations responsible for an alteration of the cell growth mechanisms of the affected tissues. The most frequently identified molecular anomalies fall within those of BWSp, mostly UPD(11)pat, and pathogenic variants in the PI3K/AKT/mTOR signaling pathway. Recently, high depth NGS approaches have increased dramatically the knowledge on the molecular defects causing these disorders, allowing classifying many of them into groups based on the cellular signal pathway involved. Indeed, when a specific molecular anomaly is found, ILO can be actually reclassified as a mild phenotype of the corresponding overgrowth disorder. However, currently few studies have evaluated what percentage of ILO cases is attributable to these specific genetic causes. Genetic testing carried out on the affected tissue (i.e., skin or muscle biopsy) rather than on leukocyte-extracted DNA allows increasing the rate of a molecular diagnosis in a relevant percentage of ILO cases [91]. Molecular tests include MS-MLPA for anomalies of the 11p15.5 region, SNP array and high-depth NGS of genes of the PI3K/AKT/mTOR and RAS/MAPK pathways; 11p15.5 methylation status testing on blood-extracted DNA allows to classify less than 6% of ILO cases as BWSp [91]. This is likely related to both a mosaicism level lower than the sensitivity limits of the analytical method, and to an inadequate sampling of the affected tissue or a combination of both factors. The rate of positive MS-MLPA test increased to 40% testing tissues other than blood [13]. It also appears likely that the ILO may have molecular bases still to be unraveled and require further studies on large case studies.

### 8.3. Cancer Risk and Surveillance

The identification of one of the typical molecular lesions makes it possible to include the ILO in a specific spectrum of disorders and to adopt related clinical guidelines. In the case of negativity to all molecular tests, a prudential clinical follow-up is adopted for such patients taking into consideration the possibility of low-expression forms of the molecular anomalies of the BWSp (i.e., the form with the highest oncogenic risk) and the related oncological implications. However, the literature indicates that ILO has a lower tumor risk than that of BWSp, with a risk ranging from 1.1% to 6% [91,92,93,94].

## 9. Conclusions

Lateralized overgrowth diagnostics is complex and laborious and takes into account a large group of genetic overgrowth conditions including BWSp, PROS/AROS, PHTS, mosaic RASopathies, and disorders of angiogenetic pathways. The physical examination allows the selection of the correct first-tier molecular test. This generally should be performed on DNA extracted from overgrown tissues and frequently requires the use of high-depth NGS techniques to detect low tissue mosaicism. Molecular diagnosis is paramount for correct management and follow-up as well as for estimation of the oncological risk. Most of these conditions are in fact characterized by cancer predisposition in childhood and require specific tumor screening.

## Figures and Tables

**Figure 1 cancers-13-06166-f001:**
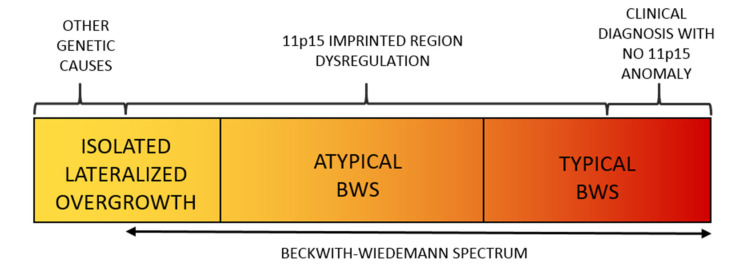
The Spectrum of the Beckwith–Wiedemann Syndrome (BWSp) includes (1) patients with “typical Beckwith–Wiedemann syndrome (BWS) with a clinical diagnosis (score ≥ 4 points) with or without an (epi)genetic change at 11p15.5 imprinted region, (2) patients with “atypical forms” (defined as those with a score < 4 points) *plus* an (epi)genetic change at the BWS locus, and (3) patients with isolated lateralized overgrowth (ILO) plus an (epi)genetic change at the 11p15.5 locus.

**Figure 2 cancers-13-06166-f002:**
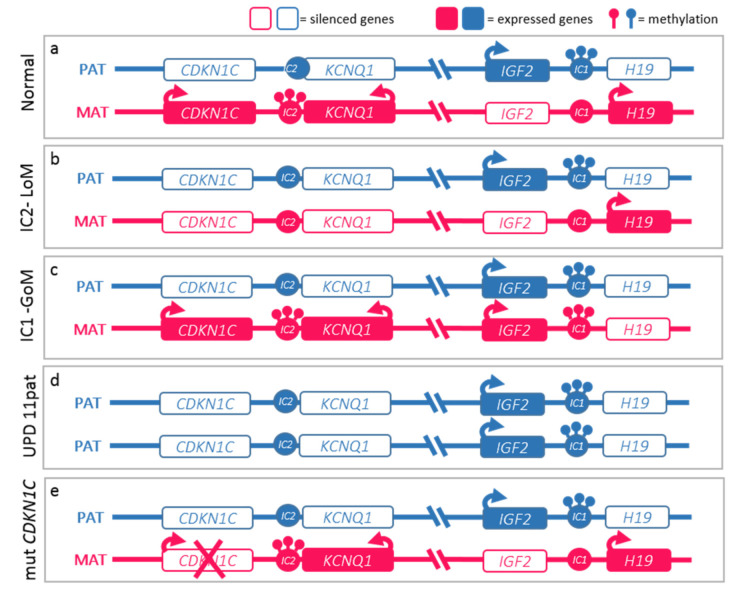
Molecular defects in Beckwith–Wiedemann spectrum (BWSp). (**a**) Normal functioning of the paternal (blue) and maternal (red) 11p15 imprinted domains. The centromeric domain is regulated by the imprinting center 2 (IC2) which methylation regulates the expression of *KCNQ1*, *KCNQ1OT1* (*KCNQ1 Opposite Strand/Antisense Transcript 1*), and *CDKN1C* expression. Normally, *KCNQ1* and *CDKN1C* are expressed by the maternal alleles and silenced on the paternal ones, while *KCNQ1OT1* is expressed only by the paternal one. The telomeric domain is regulated by the IC1: methylation on the paternal chromosome silences *H19* and allows *IGF2* expression. BWSp can be caused by: (**b**) IC2 loss of methylation (IC2-LoM) on maternal chromosome leading to reduced expression *CDKN1C*; (**c**) gain of methylation at maternal IC1 (IC1-GoM) leading to biallelic *IGF2* and reduced *H19* expression; (**d**) chromosome 11 paternal uniparental disomy (UPD(11)pat), leading to both expression anomalies seen in IC2-LoM and IC1-GoM; (**e**) *CDKN1C* loss-of-function mutations on the maternal chromosome. The latter are dominant and lead to BWSp phenotype only when inherited from the mother.

**Figure 3 cancers-13-06166-f003:**
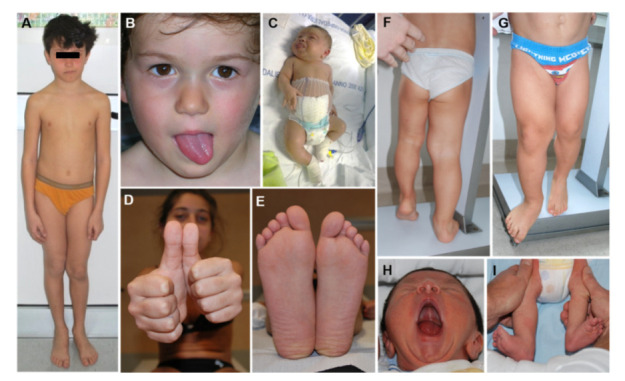
Lateralized Overgrowth (LO) in patients within the Beckwith–Wiedemann spectrum (BWSp). Isolated LO with Imprinting Center 2 Loss of Methylation (IC2-LoM, (**A**)), paternal uniparental disomy of chromosome 11 (UPD(11)pat) (**B**,**H**,**I**), and IC1 Gain of Methylation (IC1-GoM, (**D**,**E**)); severe LO in patients with BWSp clinical criteria and UPD(11)pat (**C**,**F**,**G**).

**Figure 4 cancers-13-06166-f004:**
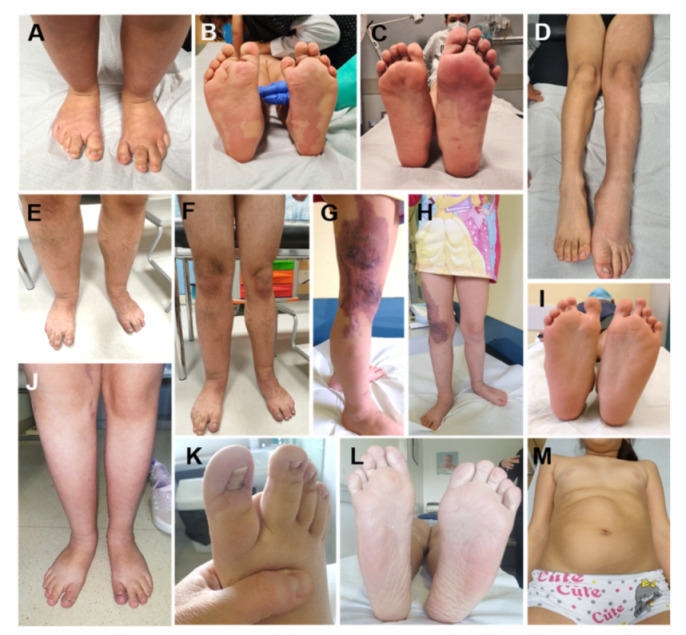
Lateralized Overgrowth (LO) in patients within the *PIK3CA*-related Overgrowth Spectrum (PROS). The latter include cases with megalencephaly-capillary malformation syndrome (MCAP, (**A**,**B**)), fibroadipose overgrowth with vascular anomalies (FAVA, (**C**,**D**,**J**–**L**)), Klippel-Trenaunay syndrome (KTS, (**E**–**H**)), congenital lipomatous overgrowth, vascular malformations, epidermal nevi, scoliosis/skeletal and spinal syndrome (CLOVES, (**I**)), hemihyperplasia multiple lipomatosis (HHML, (**M**)).

**Table 1 cancers-13-06166-t001:** Cardinal and suggestive clinical features of Beckwith–Wiedemann spectrum (BWSp). Features of BWSp are classified into cardinal ones (2 points each) and suggestive ones (1 point each). Molecular testing is indicated with ≥2 points or positive family history and inheritable 11p15 anomaly. The clinical diagnosis of BWSp can be made in cases with ≥4 points. From Brioude et al. [12].

Cardinal Features (2 Points Each)	Suggestive Features (1 Point Each)
Macroglossia	Macrosomia (height/Birth Weight > +2SD)
Exomphalos	Facial naevus simplex
Lateralized overgrowth	Polyhydramnios/Placentomegaly
Multifocal/bilateral Wilms tumor or Nephroblastomatosis	Ear creases/pits
Hyperinsulinism	Transient hypoglycemia/hyperinsulinism
Pathology findings:	Typical BWSp tumors (neuroblastoma, rhabdomyosarcoma, unilateral WT, hepatoblastoma, adrenocortical carcinoma, phaeochromocytoma)
Adrenal cortex cytomegaly	Nephromegaly/Hepatomegaly
Placental mesenchymal dysplasia	Umbilical hernia/Diastasis recti
Pancreatic adenomatosis	

## Data Availability

No new data were created or analyzed in this study. Data sharing is not applicable to this article.

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
