# Peer review of "Lateralized and Segmental Overgrowth in Children"

_cancers, 2021, doi:10.3390/cancers13246166_

Round 1

Reviewer 1 Report

Thank you for the opportunity to review the manuscript “Lateralized and Segmental Overgrowth Disorders in Children” by Alessandro Mussa et al.

Beckwith-Wiedemann Syndrome (BWS) and other overgrowth disorders are rare, genetically and clinically heterogeneous diseases. Moreover, there is a significant cancer risk and therefore it is important for the clinicians to be familiar with the possibility of these diagnoses.

Comments and suggestions

  1. Although the review is valuable in my opinion it is too long and too detailed. I would recommend to highlight the most recent findings and the most important points from the clinical point of view.
  2. Could the Authors comment on other disorders that are connected with body asymmetry? It seems to be important for the differential diagnosis, i.e. Silver-Russell syndrome.
  3. There is lack of relatively new references: doi: 3390/genes12050706, doi: 10.3389/fped.2020.613260, doi: 10.3389/fped.2020.574857, doi.org/10.1155/2014/947451

Author Response

Reply - We thank the reviewers for his/her thoughtful revision of our paper and for the comments made, all addressed in the revised version of the manuscript. We followed the indications provided by Reviewer 1 and shortened the article overall to make it more concise, focusing on the most the most important points and recent findings.

We added a short paragraph (section 8), highlighting the differential diagnosis of ILO and Silver-Russell syndrome, as requested, and provided references for that as follows.

            “An example is represented by segmental undergrowth/hemyhypoplasia than can be seen as an isolated feature in mild forms of Silver-Russell syndrome [Wakeling EL et al, 2017]. The latter is the mirror phenotype of BWS and is caused by the opposite (epi)genetic changes of BWS [Eggermann T el al, 2009]. ILO therefore, implies the differential diagnosis with causes of isolated undergrowth of a body area [Zeschnigk M et al, 2008], as it is not easy to distinguish which of the body part is normal and which affected by an over/under-growth condition.”

Finally, the new references suggested have been added throughout the text in the appropriate points.

Reviewer 2 Report

This manuscript provided a comprehensive review on the clinical presentation and genetic etiology of multiple forms of Congenital disorders of lateralized or segmental overgrowth (LO) . The authors presented this complex and heterogenous group of rare congenital disorder and their relationship to cancer in a well organized way.  It is a interesting review to read and learn about this group of disease.

My suggestion to improve the manuscript is to include brief discussion on the progress in understanding this group of diseases using experimental models such as genetically modified cell line, mouse or stem cell/organoids models. These models will help establish causal relationship between the mutations to the disease. There are certainly some work reported in mouse models on the PI3KCA/AKT-related overgrowth spectrum (PROS/AROS). And the authors should investigate into other aspect of LO in this regard.

Author Response

Reviewer #2

This manuscript provided a comprehensive review on the clinical presentation and genetic etiology of multiple forms of Congenital disorders of lateralized or segmental overgrowth (LO). The authors presented this complex and heterogenous group of rare congenital disorder and their relationship to cancer in a well organized way. It is an interesting review to read and learn about this group of disease.

My suggestion to improve the manuscript is to include brief discussion on the progress in understanding this group of diseases using experimental models such as genetically modified cell line, mouse or stem cell/organoids models. These models will help establish causal relationship between the mutations to the disease. There are certainly some work reported in mouse models on the PI3KCA/AKT-related overgrowth spectrum (PROS/AROS). And the authors should investigate into other aspect of LO in this regard.

Reply - We would like to thank this Reviewer for his/her warm words of appreciation for our article and we are happy that he/she found the topic interesting. We gladly accepted his/her suggestion to add a short discussion on experimental and animal models. These are mostly available within the PROS and have therefore been added to this section of the manuscript.

The added paragraph is the following: “To get mechanistic understanding into PROS pathophysiology and to test treatments, faithful PROS experimental models are needed, ideally spanning along the entire mutational spectrum. Temporarily regulated cell type-specific production of mutant PIK3CA alleles in animal models, in conjunction with the use of modified human stem cells and organoids, would be a useful tool to study LO in PROS. Several mice models for preclinical research demonstrated the complexities in modeling mosaic disorders due to the heterogeneity in mutational time and location in embryos. Mice constitutively expressing mutant allele Pik3ca-H1047R (one of the most oncogenic mutations) die around E9.5–E10.5 supporting the theory that strongly activating variants are lethal in the germline status. Inducible CreER expression driven by the T-Brachyury promoter can overcome early lethality and mimic PROS: specifically, mice with different mosaicism levels developed vascular anomalies similar to those observed in human [Castillo, 2016]. Several studies modeling Pik3ca-driven brain overgrowth in mice showed the relevance of mutation timing and cellular context, confirming that brain overgrowth depends on p110a activation and that postnatal Pik3ca-E545K induction cause neurological impairment [Roy A, 2015]. Furthermore, biologic models support the existence of a threshold beyond which carcinogenic mechanisms are triggered, as observed in the dosage dependent cellular consequences of homozygous or heterozygous H1047R variant [Madsen RR, 2018 and 2020]. Interestingly, the differential engineering of isogenic human induced pluripotent stem cells with heterozygous or homozygous knocking of PIK3CA H1047R, led different cellular behavior: wt-like in heterozygous state, cancer-like (with transcriptional remodeling, loss of typical epithelial morphology, decreased differentiation, up-regulation of stemness markers) in the homozygous one.”
